# Key Area Recognition and Evaluation of Audio-Visual Landscape for Global Geoparks: A Case Study of Koktokay in China

Yiting Zhu [1], Xueru Pang [1] and Chunshan Zhou [1,2,*]

1    Key Laboratory of Sustainable Development of Xinjiang's Historical and Cultural Tourism, College of Tourism, Xinjiang University, Urumqi 830046, China; yiting@xju.edu.cn (Y.Z.); pxr@stu.xju.edu.cn (X.P.)
2    School of Geography and Planning, Sun Yat-sen University, Guangzhou 510275, China
\*    Correspondence: zhoucs@mail.sysu.edu.cn

**Abstract:** A comprehensive and scientific recognition and evaluation of landscape resources is an important prerequisite for the sustainable development of global geoparks, but the existing research lacks specific means and methods. In the case of the Koktokay Global Geopark (GGp), for example, in this study, we used GIS spatial analysis, SBE, and a questionnaire survey method to construct a comprehensive evaluation path and method for an audio-visual landscape for global geoparks and further built an audio-visual preference matrix. The research results show the following: (1) The Shenzhong Canyon scenic area has the best visual evaluation effect, whereas the Golden Triangle has the worst evaluation effect. (2) Tourists are generally satisfied with the soundscapes of the ten scenic locations in the Koktokay GGp. In addition, tourists do not think that a higher or lower volume of a soundscape would make their experience more comfortable or pleasant, so an increase in the threshold value of the sound level to a level that tourists can bear while traveling is possible. (3) The Shenzhong Canyon area is located in the key landscape area; the Aiguzi Mine and Cocosuri are located in the landscape natural development area; the No. 3 Mine Pit, Eremu Lakes, the Golden Triangle, and the Karadrola Falls are located in the landscape subpriority improvement area; and the Waterfall Fossil, Betula Forest, and Karashanger Earthquake Fault Zone are located in the landscape priority improvement area. The comprehensive audio-visual landscape evaluation method constructed in this study provides a methodological tool for evaluating other similar landscapes and provides professional guidance for the protection and development of geoparks.

**Keywords:** global geoparks; landscape recognition and evaluation; visualscape; soundscape; Koktokay

## 1. Introduction

The concept of geoparks was developed in 1996, but it was only in 2000 that the European Geopark Network (EGN) was established, and it was only in 2004 that the Global Geoparks Network (GGN) was created [1]. UNESCO officially approved the International Geoscience and Geoparks Programme (IGGP) in 2015. UNESCO Global Geoparks (GGps) are defined as "Single, unified geographical areas with international geological significance where sites and landscapes contribute to the sustainable development of local communities" [2,3]. As a relatively recent form of tourism, geotourism has been consolidated with the establishment of the Europran and GGp Networks [4]. The GGps play a variety of important roles in protecting geoheritage [5], popularizing geoscience knowledge [6], developing tourism [7], promoting local socioeconomic development [8], etc. The promotion of geoheritage, geodiversity, and geoconservation, as well as the development of geoparks, are excellent opportunities to promote sustainable development [9]. As the country with the largest number of GGps in the world and an open laboratory in geosciences, China currently has 41 GGps and 289 national geoparks [1]. With the rapid development of tourism, people have gradually shifted from pursuing material enjoyment to yearning for

a spiritual and natural environment, which has also promoted the continuous upgrading of experience. The arrival of the experiential era has led people to pursue landscape experiences. Traditional landscapes based on visual experience can no longer meet people's needs, and people are growing increasingly fond of spatial environments with multiple sensory experiences. Australia's National Landscape Program was announced in 2005. It aims to enhance the value of tourism for regional economies by dispersing visitors more widely and by channeling funding into protected-area management [10]. The protection and management of GGps require not only strengthening of the physical, chemical, and ecological protection of the environment, but also awareness of the profound changes that occur at the sensory level of tourists. In addition, it is necessary to establish public policies for the sustainability and conservation of resources. However, influenced by the traditional "visual centrism" paradigm, human perception is believed to largely rely on visual factors, with 80% of sensory experiences relying on visual stimuli [11]. Previous research has mainly focused on visual factors. The evaluation method is represented by the scenic beauty evaluation (SBE) method [12], which quantitatively analyzes the beauty of a landscape, as well as the analytic hierarchy process (AHP) [13], semantic analysis (SD) [14], and other common methods. With the development of new technologies, GIS evaluation technology has been increasingly used and combined with the SBE method, resulting in evaluation results that combine visualscape quality evaluation with the evaluator's psychological aesthetic attitude, more comprehensively reflecting the visualscape evaluation quality [15]. The multisensory characteristics of tourist destinations can provide tourists with rich sensory experiences. Scholars have compared the functions of soundscapes and visualscapes in the holistic tourist experience and found that they have different impacts on tourist cognition and emotion. The soundscape directly affects the overall satisfaction of tourists with observed visualscapes [16]. The interaction between vision and hearing is of great significance for complete local cognition [17].

However, the limitations of the existing research can be summarized as follows. First, most studies are based on a single sense, whereas few studies mix two or more senses, especially combining audio-visual factors to study geoparks. In addition, landscape perception is mostly evaluated by experts. Second, most studies use a single method. In order to deal with the limitations associated with the use of a single evaluation method, comprehensive evaluation, which mixes multiple methods and tools, has become the main trend in current research. However, what sensory dimensions can be used to comprehensively evaluate the landscape in different types of scenic locations, what method should be used for comprehensive evaluation, and other key questions have not been satisfactorily answered. Therefore, in this paper, we take the Koktokay Global Geopark (the Koktokay GGp) as a research case and combine expert assessments and non-expert (i.e., tourist) judgements to identify and evaluate its audiovisual landscape resources using a multisensory landscape identification and evaluation method.

The Koktokay GGp is China's first GGp, with seismic fault zone relics, typical mineral deposits, and mining sites constituting the main landscape. As a natural landscape area integrating mountains, waters, grasslands, rare stones, and other scenery, it is an ideal place to perceive nature, acquire geoscience knowledge, and carry out tourism activities [18]. In this study, we utilized the SBE-GIS method to comprehensively obtain the quality evaluation results of the visualscape in combination with the evaluation results of the soundscape to construct an audio-visual landscape preference matrix. Key audio-visual areas of considerable significance for improving the quality of people's sensory experiences and providing a targeted scientific basis for the landscape protection planning of GGps were identified.

## 2. Literature Review

### 2.1. Sensory Experience and Tourism Experience

Tourism, as a comprehensive consumption experience, has the characteristics of a multisensory experience [19]. Tourists perceive the external world through their sense

organs and gain sensory experience. Sensory experience is caused by one or more stimuli of the five senses, including visual, aural, olfactory, gustatory, and tactile stimuli [20]. Diversified sensory experiences affect tourists' emotions, thereby affecting their tourism experiences and decision-making behaviors [21,22].

With the increasing demand for high-quality and deep experiences from tourists, sensory experience has become the most direct form of tourism experience, attracting scholars' attention. Sensory tourism research mainly involves destination sensory marketing [23], the sensory dimension of destination brand experience [24], the sensory dimension of tourism experience [25], and food destination experience [26]. Sensory marketing appeals to consumers' senses and affects their perception, judgement, and behavior [20]. Multisensory marketing is effective for tourism destination branding, creating pleasant experiences for tourists [27].

In tourism research, the research object of sensory experience involves the five human senses, namely, vision [28], hearing [29], touch [30], smell [31], and taste [32], of which vision, as the primary sense, is the most memorable and recognizable [20]. With the onset of the experience economy era, in order to create unforgettable tourism experiences, multisensory experience has become a research hotspot [33,34]. The research method has gradually shifted from quantitative or qualitative methods to a combination of the two. Questionnaire surveys and interviews are common methods for sensory data collection.

### 2.2. Landscape Evaluation Based on Sensory Experience

In the development of tourism destinations, it is necessary to conduct a scientific evaluation of landscape resources to provide a scientific basis for their positioning, functions, and protection, in which the sensory landscape plays an important role. Scholars have conducted many studies on sensory landscapes [35,36]. In 2000, the European Landscape Convention defined a landscape as a characteristic place perceived by people and presented by natural and human factors or the interaction thereof [37]. As the most direct sensory experience for people, vision often leaves a deep impression, representing an important way for people to perceive landscapes. Research has shown that despite the dominance of vision, all perception is multisensory, and that the most promising sensory modalities to investigate in combination are sound and vision [38].

Visualscapes are an important part of landscape evaluation. Previous research has mainly revolved around three aspects. The first aspect is quality evaluation of the visualscape. In the 1960s, some countries carried out a significant number of empirical and theoretical studies concerning visualscape evaluation. The main landscape quality evaluation schools are the expert school, the psychophysical school, the cognitive school, and the empirical school, which can be divided into subjective and objective schools. Daniel [39] believes that the expert approach advocated by the objective school has mainly been applied in environmental management practice, whereas the perception-based approach of the subjective school has dominated in research.

The second aspect is visualscape preference, including the impact of landscape characteristics and individual characteristics on preference. The authors of a previous study explored the relationships between visual attributes and landscape preference, investigating multiple visual attributes and concluding that future studies on landscape preference should focus on the interactive nature of visual attributes [40]. Different groups often have different subject backgrounds and different preferences for visual landscapes [41]. Another study revealed tourists' preferences regarding landscapes and their willingness to finance the conservation of landscapes [42].

The third aspect concerns the impact evaluation of visualscapes. Prior research has mainly evaluated the visual impact of rural or urban buildings from subjective and objective perspectives [43]. With the increasing demand for evaluation precision, research technology has gradually shifted from traditional photo-based questionnaire surveys to GIS [44], remote sensing [45], augmented reality (AR) [46], and other technologies.

The concept of soundscapes was first proposed by Canadian composer R.M. Schafer in the late 1960s [47]. In 2014, the International Organization for Standardization defined a soundscape as an acoustic environment perceived, experienced, and understood by a person in context [48]. The four methods commonly used in soundscape research are sound walking, laboratory experimentation, narrative interviews, and behavioral observation [49]. Most scholars have previously studied the impact of vision or hearing on the overall quality of landscape resources separately. With the deepening of research, many scholars have focused attention on the relationship between soundscapes and visualscapes [50]. Some scholars have also studied the relationships between soundscape and landscape visitor experiences. For example, by studying the impact of soundscapes on the public visitors' experiences in city parks, researchers found that music-related sounds led to positive visitor experiences, whereas mechanical sounds and traffic sounds showed negative effects [51]. In addition, many scholars have focused their attention on the impact of soundscape on tourists' behaviors and attitudes [52,53].

People are increasingly concerned about how to use different combinations of auditory and visual scenes to effectively improve personal sound perception [54]. Consistency between a soundscape and a visualscape usually enhances people's experience of a place [55]. The main research objects of audiovisual interaction include urban parks [56], urban centers [57], and residential areas [58], but few studies have focused on geoparks [59]. With the rapid development of geotourism, various artificial voices have been introduced and the geoheritage landscape has been destroyed, which has affected tourists' experience and reduced the value of landscape resources, affecting the healthy development of geoparks.

## 3. Study Design

### 3.1. Study Area and Data

The Koktokay GGp is located in Fuyun County and Qinghe County, Altay Region, northern Xinjiang Uygur Autonomous Region (Figure 1). It is 30 km away from National Highway 216 to the south, 53 km away from Fuyun County to the west, 290 km away from Altay City, and 528 km away from Urumqi City. The Koktokay GGp is generally composed of five major scenic areas, including Saihengbulak, the No. 3 Mine Pit, Erkis Grand Canyon, Cocosuri, and Karashanger. It was officially approved as a GGp in 2017. The Koktokay GGp includes mountain scenery, waterscapes, grassland, strange rocks, hot springs, and other landscapes, with diverse geoheritage and landforms as well as rich visualscape resources. The geopark also includes Erkis River, Eremu Lakes, and Cocosuri, with desirable hydrological conditions, diverse vegetation types, and abundant wildlife resources, as well as diverse soundscape resources. Therefore, the Koktokay GGp is a typical research area worth exploring.

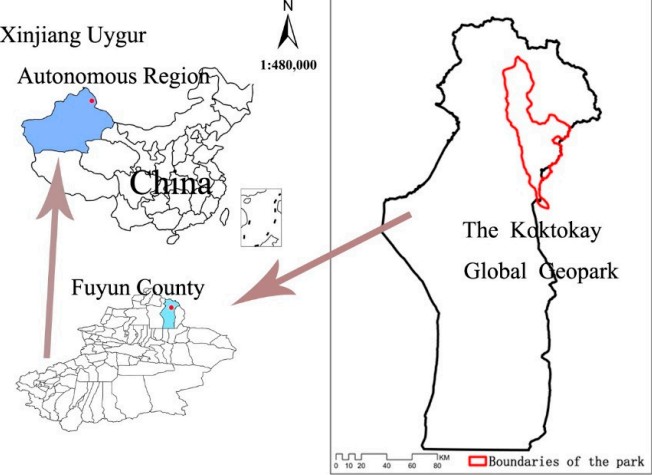

**Figure 1.** Koktokay GGp location.

According to the content of the research, regarding the specific situation of landscape resources in the geopark and discussion with relevant experts and staff familiar with the geopark, ten typical scenic locations that best represent various types of landscape resources of the geopark were selected, as shown in Figure 2: Karashanger Earthquake Fault Zone (1), Cocosuri (2), Eremu Lakes (3), the No. 3 Mine Pit (4), Aiguzi Mine (5), Shenzhong Canyon (6), Betula Forest (7), the Golden Triangle (8), Waterfall Fossil (9), and Karadrola Falls (10). These sites correspond to geosites of the geopark and are already inventoried resources. The types of landscape resources for these scenic locations are shown in Table 1.

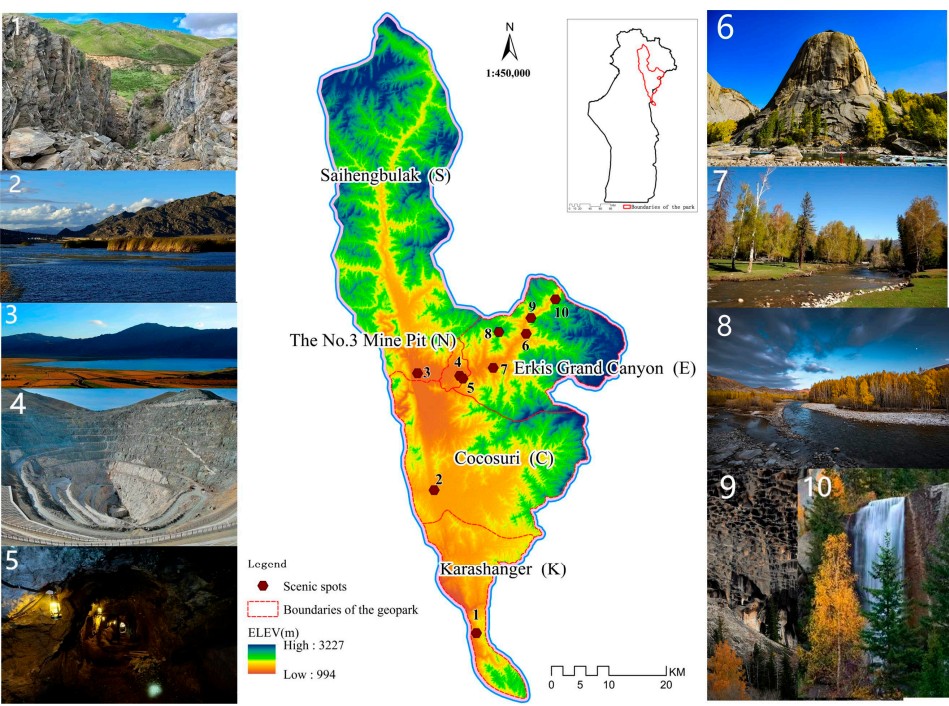

**Figure 2.** Distribution of ten typical scenic locations in the study area.

**Table 1.** Types of scenic spots and typical scenic locations.

| Landscape Resource Types | Typical Scenic Locations |
| --- | --- |
| Environmental geological landscapes | Karashanger Earthquake Fault Zone, the No. 3 Mine Pit, Aiguzi Mine |
| Environmental geological and geomorphic landscapes | Waterfall Fossil |
| Geomorphic landscapes | Shenzhong Canyon, the Golden Triangle |
| Environmental geological and water landscapes | Eremu Lakes and Karadrola Falls |
| Biological landscapes | Betula Forest |
| Water and biological landscapes | Cocosuri |

### 3.2. Study Methods

3.2.1. SBE Method

(1) Sample acquisition and evaluator identification

The survey was conducted from 28 July 2018 to 11 August 2018. Ten scenic locations were studied, and a Canon EOS800D digital camera with 24.2 million pixels was used to take photographs. All photographs were taken from 10:00 am to 5:00 pm, when the geopark had sufficient light and high visibility. At least ten photos were taken for each scenic location, for a total of 241 photos. After sorting and screening, 60 photos that best reflect the landscape characteristics of the ten scenic locations were selected for use in the subsequent evaluation process.

To ensure the objectivity and accuracy of the evaluation results, we used an evaluation group of 50 people, including university professors, students, and specialists from relevant majors, such as landscaping, geographic science, and tourism management. Among them, there were 21 males and 29 females (Table 2).

**Table 2.** Composition of the evaluator group.

| Category | Participant | Number |
|---|---|---|
| Expert group | Professors, associate professors, lecturers, engineers | 13 |
| Student group | Undergraduates, postgraduates, and doctoral students | 37 |

(2) Evaluation Methodology

Visual landscape evaluation of the Koktokay GGp was carried out using the SBE method, which has been widely used in psychophysics. A total of 60 photos were converted into slides, randomly numbered, and displayed at intervals of ten seconds. A five-point Likert scale was used to score the various evaluation indicators of scenic beauty.

The *SBE* score calculation formula was as follows [60]:

$$SBE = \frac{(s_{11} + s_{12} + \ldots + s_{1n}) + (s_{21} + s_{22} + \ldots + s_{2n}) + \ldots + (s_{i1} + s_{i2} + \ldots + s_{in})}{i \times n} \quad (1)$$

where $S_{in}$ is the *SBE* score of the *i*th scenic location assigned by the nth evaluator (*i* = 1, 2, 3, ..., 10) and *n* is the number of evaluators (*n* = 1, 2, 3, ..., 50).

In addition, the evaluation results need to be standardized to eliminate the differences caused by different aesthetic attitudes of the evaluators.

### 3.2.2. GIS Spatial Analysis

Landscape visual sensitivity was used to indicate the degree to which the landscape was noticed by visitors during the tour. Based on the relevant literature [61], discussion with experts, and the actual situation of the geopark, the three factors of relative slope ($S_a$), relative distance ($S_d$), and sight probability ($S_t$) were selected as the evaluation factors for landscape visual sensitivity. The evaluation criteria are listed in Table 3.

**Table 3.** Evaluation criteria for landscape visual sensitivity.

| Landscape Visual Sensitivity | Score | | | |
|---|---|---|---|---|
| | 1 | 2 | 3 | 4 |
| Relative slope ($S_a$, °) | $S_a \leq 15°$ (Low-sensitivity zone) | $15° < S_a \leq 30°$ (Moderate-sensitivity zone) | $30° < S_a \leq 90°$ (High-sensitivity zone) | |
| Relative distance ($S_d$, m) | $S_d > 1500$ m (Rarely seen zone) | $800 < S_d \leq 1500$ (Distant view zone) | $400 < S_d \leq 800$ (Mid-shot zone) | $0 < S_d \leq 400$ (Close-shot zone) |
| Sight probability ($S_t$) | $0 < S_t \leq 10$ | $10 < S_t \leq 20$ | $20 < S_t \leq 30$ | |

### 3.2.3. Soundscape Evaluation Method

A field survey was conducted at ten typical scenic locations in the Koktokay GGp to measure the physical sound level. Questionnaires were used to investigate tourists' feelings about the sound volume at these scenic locations. The survey period lasted from July 28 to 11 August 2018.

A sound level meter was used to obtain the corresponding data at the selected locations, and the physical sound level values of ten scenic locations were calculated according to

Formula (2) [62]. Then, the average value of the sound level was used as the volume reference value for the scenic locations.

$$L_{eq} = 10\lg\left(\frac{1}{T}\int_0^T 10^{0.1L_A}dt\right) \qquad (2)$$

where $L_{eq}$ is the equivalent continuous A-weighted sound level, which refers to the average sound level within the specified measurement time ($T$), and $L_A$ is the instantaneous sound level at moment $t$.

A questionnaire survey was completed by tourists at the scenic locations where sound levels were measured in order to understand their satisfaction with the soundscape. In the questionnaire design, a 5-point Likert scale was used to express the degree of satisfaction of the tourists with the soundscape. The overall satisfaction of tourists with the soundscape resources of the scenic location where they were located at the time was rated to varying degrees, i.e., "very satisfied", "relatively satisfied", "basically satisfied", "dissatisfied", and "very dissatisfied". Scores ranged from 5 to 1, and Formula (3) was used to calculate the tourists' satisfaction with the soundscape [62].

$$M = \sum_{i=1}^{10} \frac{n_i}{N} m_i \qquad (3)$$

where $M$ represents the satisfaction of tourists with the soundscape, $m_i$ refers to the tourist rating of ten scenic locations on a Likert scale, $n_i$ represents the total number of people who gave the same score, and $N$ represents the total number of people who completed the questionnaire.

3.2.4. Importance–Performance Analysis (IPA)

IPA analyzes quality attributes in two dimensions—performance and importance—then integrates these two aspects into a matrix to guide a company or destination in choosing the most appropriate strategy to improve competitiveness [63]. In this study, we applied the main ideas of the IPA method to landscape evaluation. Taking (0,0) as the coordinate origin, with the visual evaluation of landscape resources as the horizontal axis and the soundscape evaluation as the vertical axis, an audio-visual preference matrix was constructed and specifically divided into four quadrants.

**4. Research Results**

*4.1. Visual Landscape Identification and Evaluation*

The scores of the 50 professional evaluators were calculated to derive the SBE score and the standardized score of each scenic location, as follows (Table 4). The results show that the SBE scores of all ten scenic locations were greater than 2, with an average SBE score of 3.329, which indicates that the landscape resources of the Koktokay GGp are of a high quality in terms of beauty.

**Table 4.** The SBE scores of the scenic locations.

| Scenic Location | SBE Score | Standardized Value |
|---|---|---|
| Karashanger Earthquake Fault Zone | 2.644 | −0.647 |
| Cocosuri | 4.161 | 0.446 |
| Eremu Lakes | 3.637 | 0.313 |
| The No. 3 Mine Pit | 3.023 | −0.462 |
| Aiguzi Mine | 3.522 | 0.294 |
| Shenzhong Canyon | 4.347 | 0.580 |
| Betula Forest | 4.248 | 0.172 |
| Golden Triangle | 3.116 | −0.281 |
| Waterfall Fossil | 4.028 | 0.362 |
| Karadrola Falls | 3.082 | −0.518 |

The SBE score of each scenic location was determined as follows (in descending order): Shenzhong Canyon > Betula Forest > Cocosuri > Waterfall Fossil > Eremu Lakes > Aiguzi Mine > Golden Triangle > Karadrola Falls > The No. 3 Mine Pit > Karashanger Earthquake Fault Zone. Shenzhong Canyon had the highest score, with a standardized score of 0.580. Shenzhong Canyon is the most iconic geoheritage landscape of the Koktokay GGp, located in the middle of the granite canyon tourist area in Erkis Grand Canyon, with high aesthetic value. Cocosuri has a standardized score of 0.446, second only to Shenzhong Canyon, and because of its rich aquatic life and beautiful scenery, it is a transit station for Kazakh herders' transhumance and an excellent place for viewing and photographing grasslands, wetlands, folded gullies, and various kinds of perching birds. Among the two most famous sites in the Koktokay GGp, the No. 3 Mine Pit ranks second to last, with a standardized value of −0.612, and its SBE score is lower than that of the Aiguzi Mine. The reason for such a low ranking is that, although the No. 3 Mine Pit is much more famous than the Aiguzi Mine, it has a single type of tour and only serves as a photo location for tourists, whereas Aiguzi Mine includes re-creations of mining scenes and a wide variety of ores, providing considerable educational value. The standardized value of the Karashanger Earthquake Fault Zone is −0.647, ranking last because the SBE of the attractions is based entirely on geological phenomena.

In this study, ArcGIS10.0 software was used to extract the relative slope from the digital elevation model (DEM) of the Koktokay GGp, and the sensitivity of the relative slope was divided into three levels after classification by the natural interval method. Next, the viewable range of each site and the sensitivity zone of the relative slope were superimposed (Figure 3).

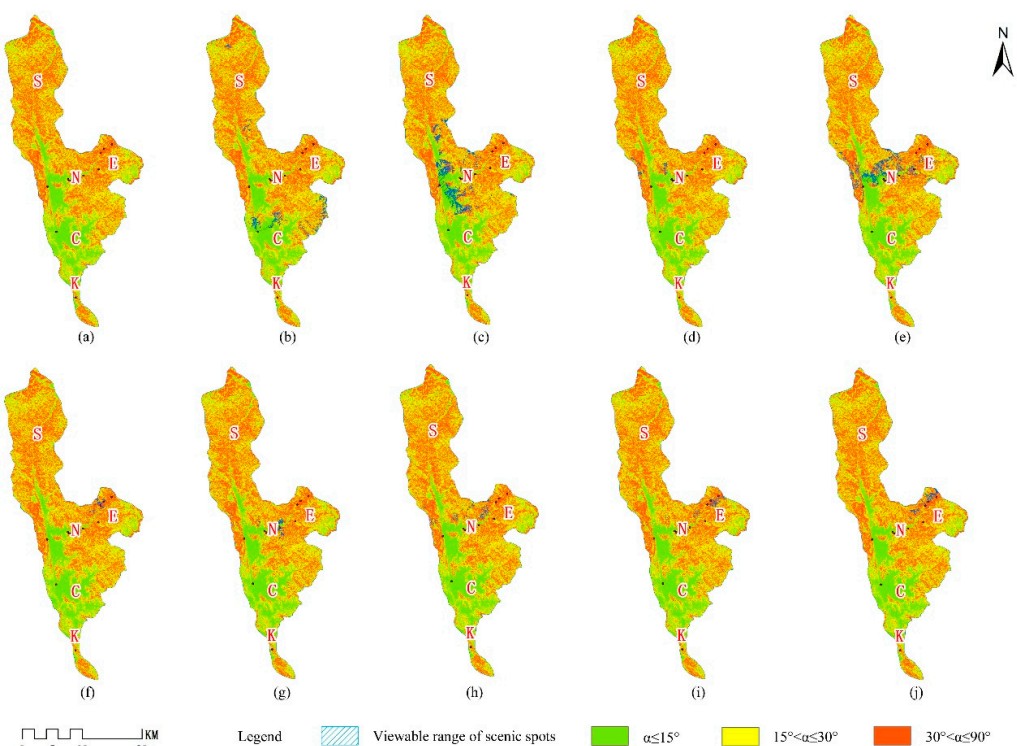

**Figure 3.** Relative slope sensitivity of the viewable range of each scenic location. (**a–j**) Karashanger Earthquake Fault Zone, Cocosuri, Eremu Lakes, The No. 3 Mine Pit, Aiguzi Mine, Shenzhong Canyon, Betula Forest, Golden Triangle, Waterfall Fossil, and Karadrola Falls. Five major scenic areas, including Saihengbulak (S), The No. 3 Mine Pit (N), Erkis Grand Canyon (E), Cocosuri (C), and Karashanger (K).

The overlap area was calculated, and corresponding scores were assigned according to the proportion of the viewable range of each sensitivity zone to the total area of viewable

range and the evaluation criteria listed in Table 3; the results are shown in Table 5. The total score of relative slope sensitivity was determined for each scenic location as follows (in descending order): Waterfall Fossil > Karashanger Earthquake Fault Zone > Shenzhong Canyon > Betula Forest > Karadrola Falls > The No. 3 Mine Pit > Golden Triangle > Cocosuri > Aiguzi Mine > Eremu Lakes.

**Table 5.** Evaluation of relative slope sensitivity.

| Scenic Location | Sensitivity Zone of Relative Slope | Area of Viewable Range (km$^2$) | Proportion of the Viewable Range | Corresponding Score | Total Score |
|---|---|---|---|---|---|
| Karashanger Earthquake Fault Zone | Low | 0.012 | 0.060 | 0.060 | |
| | Moderate | 0.044 | 0.223 | 0.446 | 2.656 |
| | High | 0.142 | 0.717 | 2.150 | |
| Cocosuri | Low | 5.154 | 0.175 | 0.175 | |
| | Moderate | 15.543 | 0.528 | 1.056 | 2.122 |
| | High | 8.747 | 0.297 | 0.891 | |
| Eremu Lakes | Low | 23.082 | 0.384 | 0.384 | |
| | Moderate | 19.370 | 0.322 | 0.645 | 1.909 |
| | High | 17.630 | 0.293 | 0.880 | |
| The No. 3 Mine Pit | Low | 0.400 | 0.077 | 0.077 | |
| | Moderate | 2.096 | 0.406 | 0.812 | 2.439 |
| | High | 2.664 | 0.516 | 1.549 | |
| Aiguzi Mine | Low | 20.271 | 0.272 | 0.272 | |
| | Moderate | 25.778 | 0.346 | 0.693 | 2.109 |
| | High | 28.378 | 0.381 | 1.144 | |
| Shenzhong Canyon | Low | 0.170 | 0.045 | 0.045 | |
| | Moderate | 0.975 | 0.261 | 0.521 | 2.649 |
| | High | 2.596 | 0.694 | 2.082 | |
| Betula Forest | Low | 0.196 | 0.043 | 0.043 | |
| | Moderate | 1.410 | 0.310 | 0.620 | 2.604 |
| | High | 2.942 | 0.647 | 1.940 | |
| Golden Triangle | Low | 1.409 | 0.089 | 0.089 | |
| | Moderate | 6.449 | 0.407 | 0.815 | 2.415 |
| | High | 7.976 | 0.504 | 1.511 | |
| Waterfall Fossil | Low | 0.193 | 0.038 | 0.038 | |
| | Moderate | 0.993 | 0.193 | 0.387 | 2.731 |
| | High | 3.946 | 0.769 | 2.307 | |
| Karadrola Falls | Low | 0.484 | 0.059 | 0.059 | |
| | Moderate | 2.910 | 0.352 | 0.705 | 2.530 |
| | High | 4.864 | 0.589 | 1.767 | |

With the help of ArcGIS10.0 software, a rank distribution map of relative distance sensitivity was generated, and the viewable range of each scenic location was overlaid with the sensitivity zone of the relative distance (Figure 4). The overlap area was calculated, and the corresponding scores were assigned according to the proportion of the viewable range of each sensitivity zone to the total area of viewable range and the evaluation criteria listed in Table 3. The total scores of relative distance sensitivity were determined for the ten scenic locations as follows (in descending order; Table 6): Karashanger Earthquake Fault Zone > Shenzhong Canyon > Betula Forest > Waterfall Fossil > Aiguzi Mine > Karadrola Falls > Eremu Lakes > The No. 3 Mine Pit > Golden Triangle > Cocosuri.

With the help of ArcGIS10.0 software, the viewable ranges of the ten scenic locations were overlaid with the sensitivity zones of sight probability (Figure 5). The overlap area was calculated, and the corresponding scores were assigned according to the proportion of the viewable range of each sensitive zone to the total area of the viewable range, as well as the evaluation criteria listed in Table 3. The evaluation results of the sight probability sensitivity for each scenic location were determined as follows (in descending order; Table 7): Cocosuri > The No. 3 Mine Pit > Eremu Lakes > Aiguzi Mine > Shenzhong Canyon > Waterfall Fossil > Golden Triangle > Betula Forest > Karadrola Falls > Karashanger Earthquake Fault Zone.

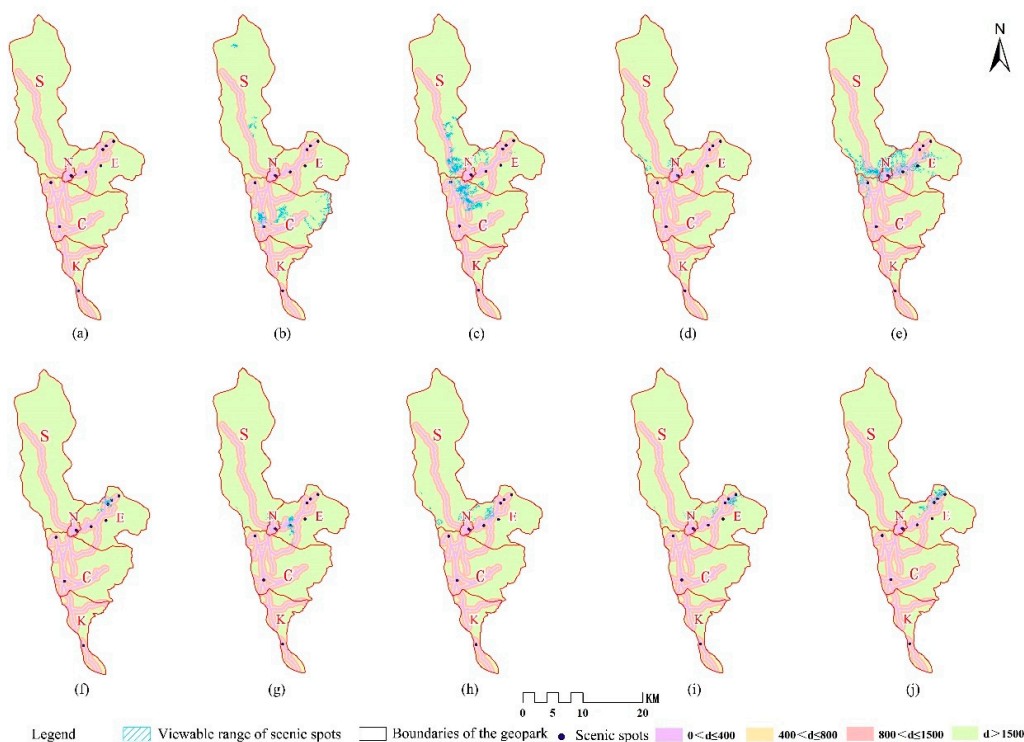

**Figure 4.** Relative distance sensitivity of the viewable range of each scenic location. (**a**–**j**) Karashanger Earthquake Fault Zone, Cocosuri, Eremu Lakes, The No. 3 Mine Pit, Aiguzi Mine, Shenzhong Canyon, Betula Forest, Golden Triangle, Waterfall Fossil, and Karadrola Falls. Five major scenic areas, including Saihengbulak (S), The No. 3 Mine Pit (N), Erkis Grand Canyon (E), Cocosuri (C), and Karashanger (K).

**Table 6.** Evaluation of relative distance sensitivity.

| Scenic Location | Sensitivity Zone of Relative Distance | Area of Viewable Range (km²) | Proportion of Viewable Range | Corresponding Score | Total Score |
|---|---|---|---|---|---|
| Karashanger Earthquake Fault Zone | Close-shot zone | 0.138 | 0.698 | 2.792 | |
| | Mid-shot zone | 0.054 | 0.274 | 0.823 | |
| | Distant view zone | 0.030 | 0.028 | 0.055 | 3.671 |
| | Rarely seen zone | 0 | 0.000 | 0.000 | |
| Cocosuri | Close-shot zone | 0.914 | 0.031 | 0.124 | |
| | Mid-shot zone | 0.960 | 0.033 | 0.098 | |
| | Distant view zone | 1.124 | 0.038 | 0.076 | 1.196 |
| | Rarely seen zone | 26.447 | 0.898 | 0.898 | |
| Eremu Lakes | Close-shot zone | 5.218 | 0.087 | 0.347 | |
| | Mid-shot zone | 6.487 | 0.108 | 0.324 | |
| | Distant view zone | 13.343 | 0.222 | 0.444 | 1.699 |
| | Rarely seen zone | 35.034 | 0.583 | 0.583 | |
| The No. 3 Mine Pit | Close-shot zone | 0.382 | 0.074 | 0.296 | |
| | Mid-shot zone | 0.359 | 0.070 | 0.209 | |
| | Distant view zone | 0.913 | 0.177 | 0.354 | 1.538 |
| | Rarely seen zone | 3.505 | 0.679 | 0.679 | |
| Aiguzi Mine | Close-shot zone | 9.019 | 0.121 | 0.485 | |
| | Mid-shot zone | 10.806 | 0.145 | 0.436 | |
| | Distant view zone | 13.436 | 0.181 | 0.361 | 1.834 |
| | Rarely seen zone | 41.167 | 0.553 | 0.553 | |
| Shenzhong Canyon | Close-shot zone | 1.477 | 0.395 | 1.579 | |
| | Mid-shot zone | 1.049 | 0.280 | 0.841 | |
| | Distant view zone | 1.033 | 0.276 | 0.552 | 3.021 |
| | Rarely seen zone | 0.182 | 0.049 | 0.049 | |

**Table 6.** *Cont.*

| Scenic Location | Sensitivity Zone of Relative Distance | Area of Viewable Range (km²) | Proportion of Viewable Range | Corresponding Score | Total Score |
|---|---|---|---|---|---|
| Betula Forest | Close-shot zone | 0.749 | 0.165 | 0.659 | |
| | Mid-shot zone | 1.394 | 0.306 | 0.919 | 2.475 |
| | Distant view zone | 1.672 | 0.368 | 0.735 | |
| | Rarely seen zone | 0.733 | 0.161 | 0.161 | |
| Golden Triangle | Close-shot zone | 0.671 | 0.042 | 0.170 | |
| | Mid-shot zone | 1.128 | 0.071 | 0.214 | 1.461 |
| | Distant view zone | 3.030 | 0.191 | 0.383 | |
| | Rarely seen zone | 11.005 | 0.695 | 0.695 | |
| Waterfall Fossil | Close-shot zone | 0.669 | 0.130 | 0.521 | |
| | Mid-shot zone | 1.230 | 0.240 | 0.719 | 2.218 |
| | Distant view zone | 1.782 | 0.347 | 0.695 | |
| | Rarely seen zone | 1.450 | 0.283 | 0.283 | |
| Karadrola Falls | Close-shot zone | 0.706 | 0.086 | 0.342 | |
| | Mid-shot zone | 0.920 | 0.111 | 0.334 | 1.779 |
| | Distant view zone | 2.475 | 0.300 | 0.599 | |
| | Rarely seen zone | 4.156 | 0.503 | 0.503 | |

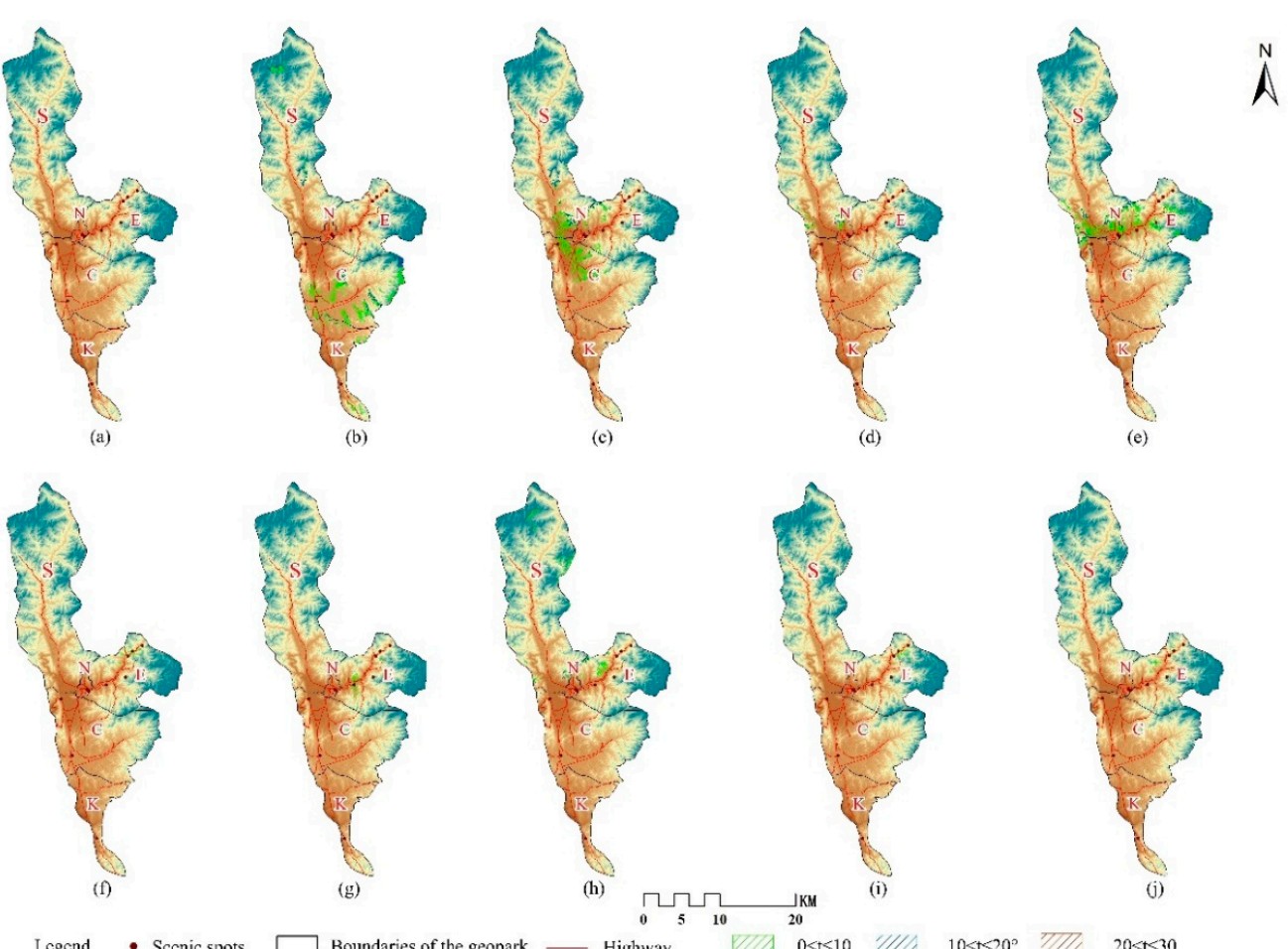

**Figure 5.** Sight probability sensitivity of the viewable range of each scenic location. (**a–j**) Karashanger Earthquake Fault Zone, Cocosuri, Eremu Lakes, The No. 3 Mine Pit, Aiguzi Mine, Shenzhong Canyon, Betula Forest, Golden Triangle, Waterfall Fossil, and Karadrola Falls. Five major scenic areas, including Saihengbulak (S), The No. 3 Mine Pit (N), Erkis Grand Canyon (E), Cocosuri (C), and Karashanger (K).

**Table 7.** Evaluation of sight probability sensitivity.

| Scenic Location | Sight Probability | Area of Viewable Range (km²) | Proportion of the Viewable Range | Corresponding Score | Total Score |
|---|---|---|---|---|---|
| Karashanger Earthquake Fault Zone | $0 < S_t \leq 10$ | 0.186 | 1 | 1 | |
| | $10 < S_t \leq 20$ | 0 | 0 | 0 | 1 |
| | $20 < S_t \leq 30$ | 0 | 0 | 0 | |
| Cocosuri | $0 < S_t \leq 10$ | 48.003 | 0.761 | 0.761 | |
| | $10 < S_t \leq 20$ | 15.065 | 0.239 | 0.477 | 1.240 |
| | $20 < S_t \leq 30$ | 0.049 | 0.001 | 0.002 | |
| Eremu Lakes | $0 < S_t \leq 10$ | 53.955 | 0.839 | 0.839 | |
| | $10 < S_t \leq 20$ | 10.308 | 0.160 | 0.321 | 1.162 |
| | $20 < S_t \leq 30$ | 0.056 | 0.001 | 0.003 | |
| The No. 3 Mine Pit | $0 < S_t \leq 10$ | 6.361 | 0.763 | 0.763 | |
| | $10 < S_t \leq 20$ | 1.974 | 0.237 | 0.473 | 1.239 |
| | $20 < S_t \leq 30$ | 0.005 | 0.001 | 0.003 | |
| Aiguzi Mine | $0 < S_t \leq 10$ | 61.163 | 0.879 | 0.879 | |
| | $10 < S_t \leq 20$ | 8.265 | 0.119 | 0.238 | 1.123 |
| | $20 < S_t \leq 30$ | 0.144 | 0.002 | 0.006 | |
| Shenzhong Canyon | $0 < S_t \leq 10$ | 5.442 | 0.945 | 0.945 | |
| | $10 < S_t \leq 20$ | 0.318 | 0.055 | 0.110 | 1.055 |
| | $20 < S_t \leq 30$ | 0 | 0 | 0 | |
| Betula Forest | $0 < S_t \leq 10$ | 6.083 | 0.998 | 0.998 | |
| | $10 < S_t \leq 20$ | 0.024 | 0.004 | 0.008 | 1.006 |
| | $20 < S_t \leq 30$ | 0 | 0 | 0 | |
| Golden Triangle | $0 < S_t \leq 10$ | 22.834 | 0.966 | 0.966 | |
| | $10 < S_t \leq 20$ | 0.775 | 0.033 | 0.066 | 1.035 |
| | $20 < S_t \leq 30$ | 0.020 | 0.001 | 0.003 | |
| Waterfall Fossil | $0 < S_t \leq 10$ | 4.784 | 0.957 | 0.957 | |
| | $10 < S_t \leq 20$ | 0.214 | 0.043 | 0.086 | 1.043 |
| | $20 < S_t \leq 30$ | 0 | 0 | 0 | |
| Karadrola Falls | $0 < S_t \leq 10$ | 7.423 | 0.997 | 0.997 | |
| | $10 < S_t \leq 20$ | 0.019 | 0.003 | 0.005 | 1.003 |
| | $20 < S_t \leq 30$ | 0 | 0 | 0 | |

After calculating the total landscape sensitivity scores of ten typical scenic locations in the Koktokay GGp, the GIS and SBE evaluation results were weighted to determine comprehensive visual evaluation scores for the landscapes as follows (in descending order): Shenzhong Canyon > Karashanger Earthquake Fault Zone > Waterfall Fossil > Betula Forest > Aiguzi Mine > Eremu Lakes > Cocosuri > Karadrola Falls > The No. 3 Mine Pit > Golden Triangle (Table 8).

**Table 8.** Comprehensive visual evaluation of the landscape at each scenic location.

| Scenic Location | Landscape Visual Sensitivity Score | | | Total Landscape Sensitivity Score | Standardized Landscape Sensitivity Score | Standardized SBE Score | Weighted Score | Ranking |
|---|---|---|---|---|---|---|---|---|
| | Relative Slope | Relative Distance | Sight Probability | | | | | |
| Karashanger Earthquake Fault Zone | 2.656 | 3.671 | 1.000 | 7.327 | 1.909 | −0.647 | 1.262 | 2 |
| Cocosuri | 2.122 | 1.196 | 1.240 | 4.559 | −1.144 | 0.446 | −0.698 | 7 |
| Eremu Lakes | 1.909 | 1.699 | 1.162 | 4.770 | −0.911 | 0.313 | −0.598 | 6 |
| The No. 3 Mine Pit | 2.439 | 1.538 | 1.239 | 5.216 | −0.419 | −0.612 | −1.031 | 9 |
| Aiguzi Mine | 2.109 | 1.834 | 1.123 | 5.066 | −0.585 | 0.294 | −0.291 | 5 |
| Shenzhong Canyon | 2.649 | 3.021 | 1.055 | 6.725 | 1.245 | 0.580 | 1.825 | 1 |
| Betula Forest | 2.604 | 2.475 | 1.006 | 6.084 | 0.538 | 0.172 | 0.710 | 4 |
| Golden Triangle | 2.415 | 1.461 | 1.035 | 4.910 | −0.757 | −0.281 | −1.038 | 10 |
| Waterfall Fossil | 2.731 | 2.218 | 1.043 | 5.992 | 0.437 | 0.362 | 0.799 | 3 |
| Karadrola Falls | 2.530 | 1.779 | 1.003 | 5.312 | −0.313 | −0.518 | −0.831 | 8 |

The comprehensive scores of Shenzhong Canyon, Karashanger Earthquake Fault Zone, Waterfall Fossil, and Betula Forest were positive, which indicates that these scenic locations are the dominant attractions in the Koktokay GGp, with rich landscape visual resources. On the other hand, the scores of the remaining six scenic locations are negative, indicating that their visual landscape resources are relatively deficient and in need of development and upgrading. The geomorphic landscape of Shenzhong Canyon has the highest weighted

score. Although its visual sensitivity score is not the highest, its visual beauty is favored by tourists. The Karashanger Earthquake Fault Zone ranked second, and although it was not favored by evaluators, its landscape visual sensitivity advantage is significant. Waterfall Fossil ranked third because its poor visual sensitivity was compensated for by the evaluators' appreciation of its visual beauty. Betula Forest was ranked in fourth place, with the appreciation of the evaluators to making up for its poor visual sensitivity. The reason for the poor comprehensive visual evaluation results of the last five scenic locations was the lower scores of subjective SBE evaluation and objective GIS evaluation. Among them, Eremu Lakes and Cocosuri both belong to the category of water landscapes, with beautiful scenery and a good ecological environment, which meet the needs of tourists for leisure tourism. However, their terrains are low and the water landscapes fluctuate slightly, making GIS evaluation not advantageous. As a geomorphic landscape, Golden Triangle is in a concealed location with small slopes, and GIS evaluation is not advantageous. Karadrola Falls and the No. 3 Mine Pit belong to environmental geological landscapes. Although their slopes are large and their terrains are high and easy to identify, their aesthetic levels are not high.

*4.2. Recognition and Evaluation of Soundscape*

A field survey measuring the physical sound level was conducted at ten typical scenic locations in the Koktokay GGp from 28 July to 31 July. We randomly selected two areas with high concentrations of tourists for measurement in each scenic location. The measurement time generally lasted from 11:00 to 20:00 every day, as there are many visitors in the geopark during this period. Relatively numerous complete data can be obtained to ensure the reliability and scientificity of the results. During measurement, a sound level meter was used to read a sound level every 5 s at each measurement point, measuring 20 times and taking a total of 100 sound level readings. The average value of the sound level calculated at each measurement point of the scenic location was used as the reference value for the volume level of the scenic location, as shown in Table 9. The noise evaluation of tourist areas in China adopts the standards in Table 10, among which geoparks are Type 2 natural scenic spots, and the standard for daytime noise in scenic areas is 45 dB. There are also regulations that explicitly stipulate that the outdoor noise levels of scenic spots should be controlled below the standard established by the GB3096-93 Urban Area Environmental Noise Standard (50 dB in daytime and 40 dB at night) [62]. From the measured volume results of the scenic locations, it can be seen that the volumes of most scenic locations were higher than the noise control standards. Normally, tourists will choose to leave, but it was found in the survey that tourists did not choose to leave. Therefore, this study needed to obtain tourists' satisfaction with the soundscape of each scenic location through a questionnaire survey.

**Table 9.** The volume of tourism soundscape in ten typical scenic locations.

| Scenic Location | Volume (dB) | Scenic Location | Volume (dB) |
|---|---|---|---|
| Karashanger Earthquake Fault Zone | 43.71 | Shenzhong Canyon | 62.04 |
| Cocosuri | 56.23 | Betula Forest | 53.22 |
| Eremu Lakes | 48.81 | Golden Triangle | 55.19 |
| The No. 3 Mine Pit | 59.27 | Waterfall Fossil | 64.18 |
| Aiguzi Mine | 57.34 | Karadrola Falls | 78.36 |

**Table 10.** Standard value of environmental noise in scenic areas.

| Category | Standard Vale in Daytime | Suitable Region |
|---|---|---|
| 1 | 40 | Tourism resort district, convalescence area, Holiday villa, senior hotel zone |
| 2 | 45 | Natural scenic spot |
| 3 | 50 | City tourist area |
| 4 | 55 | Historical and cultural tourist area |
| 5 | 60 | Entertainment and sports tourist area |

The survey questionnaire was conducted at ten selected scenic locations from 1 August to 11 August 2018. A total of 1000 questionnaires were distributed—100 for each scenic location. A total of 941 valid questionnaires were completed, with a response rate of 94.1%. The questionnaires for tourists were administered using a random sampling method, allowing tourists to make judgments based on their actual feelings after sightseeing. The sample demographic characteristics are shown in Table 11. In terms of gender, roughly 61.88% of the tourists were women. In terms of age, the majority were young and middle-aged, accounting for 69.82% of the total. In terms of education level, the majority were undergraduate and vocational college students, accounting for 63.02% of the total. In terms of income, the middle-income group was the main group. In terms of occupational distribution, production, and transportation, equipment operators, professional technicians, and managers of enterprises and institutions constituted the main body. In terms of regional composition, the proportion of tourists from the Xinjiang Autonomous Region and other provinces was approximately 6:4. The tourists in Xinjiang mainly came from Yili, Urumqi, Changji Prefecture, Karamay and other surrounding areas, while tourists from other provinces mainly came from Guangdong, Henan, the Northeast, and other regions. The collected data are basically consistent with the characteristics of the local tourism market. After a statistical analysis of the valid questionnaires, the perceptions of tourists regarding soundscapes in the Koktokay GGp were obtained (Table 12). The satisfaction scores of the tourists at the ten scenic locations in the Koktokay GGp were determined as follows (in descending order): Cocosuri > Shenzhong Canyon > Aiguzi Mine > Waterfall Fossil > Betula Forest > The No. 3 Mine Pit > Eremu Lakes > Golden Triangle > Karadrola Falls > Karashanger Earthquake Fault Zone. The soundscape satisfaction scores fell in the range of 3–5 points, i.e., between basic satisfaction and relative satisfaction.

**Table 11.** Characteristics of tourist samples.

| Variable | Value | Frequency | Percent |
|---|---|---|---|
| Gender | Male | 359 | 38.12 |
| | Female | 582 | 61.88 |
| Age | ≤19 | 53 | 5.63 |
| | 20–34 | 323 | 34.33 |
| | 35–49 | 334 | 35.49 |
| | 50–64 | 168 | 17.85 |
| | ≥65 | 63 | 6.70 |
| Occupation | Managers of enterprises and institutions | 163 | 17.32 |
| | Professional and technical personnel | 229 | 24.34 |
| | Office staff and related personnel | 138 | 14.67 |
| | Production personnel in agriculture, forestry, animal husbandry, fishing, and water conservancy industries | 39 | 4.14 |
| | Production and transportation equipment operating personnel and related personnel | 237 | 25.19 |
| | Military | 36 | 3.83 |
| | Free profession | 99 | 10.52 |
| Education | Junior high school or below | 69 | 7.33 |
| | High school | 124 | 13.20 |
| | Junior college and university | 593 | 63.02 |
| | Master degree or above | 155 | 16.47 |
| Income | CNY < 2000 | 113 | 12.01 |
| | CNY 2000–4000 | 563 | 59.83 |
| | CNY 4000–6000 | 163 | 17.32 |
| | CNY > 6000 | 102 | 10.84 |
| Region | The Xinjiang Uygur Autonomous Region | 563 | 59.83 |
| | Other provinces | 378 | 40.17 |

**Table 12.** The results for the tourism soundscapes in ten typical scenic locations.

| Scenic Location | Satisfaction Score | Standardized Value | Tourist Perception of Volume |
|---|---|---|---|
| Karashanger Earthquake Fault Zone | 3.11 | −1.171 | Relatively low |
| Cocosuri | 4.29 | 1.422 | Moderate |
| Eremu Lakes | 3.38 | −0.578 | Moderate |
| The No. 3 Mine Pit | 3.40 | −0.534 | Relatively low |
| Aiguzi Mine | 4.24 | 1.312 | Relatively low |
| Shenzhong Canyon | 4.27 | 1.378 | Moderate |
| Betula Forest | 3.58 | −0.138 | Relatively low |
| Golden Triangle | 3.28 | −0.798 | Relatively low |
| Waterfall Fossil | 3.62 | −0.051 | Moderate |
| Karadrola Falls | 3.26 | −0.842 | Moderate |

Further analysis of the physical properties of soundscapes shows that tourists do not believe that the louder the better or the smaller the better in terms of soundscapes. While too much silence can cause fear, too much noise can cause people to want to leave an area. The sound perception scores of the Karashanger Earthquake Fault Zone, the No. 3 Mine Pit, Golden Triangle, and Betula Forest were low, whereas the sound perception score of Karadrola Falls was moderate, showing that the subjective perception of soundscape and the physical parameters of sound are not completely consistent with each other. People's perception of noise and the acceptable threshold are not the same in scenic locations as they are in daily urban environments. From the perspective of noise evaluation, the noise level of the ten observation points in the Koktokay GGp exceeded the standard value (50 dB) of five types of environmental noise in urban areas, a value which has been used in tourism areas, except for the Karashanger Earthquake Fault Zone and Eremu Lakes. However, tourists' perceptions and experiences of a soundscape can increase their thresholds of acceptable sound levels in the process of sightseeing. In particular, the sound of running water at three observation points, i.e., Shenzhong Canyon, Waterfall Fossil, and Karadrola Falls, was very loud, but tourists rated it as moderate. The soundscape volume values of the ten scenic locations were basically satisfactory, indicating that the soundscapes of the Koktokay GGp are all within the acceptable threshold range. A tourist's environmental experience is positively impacted if they feel comfortable with the soundscape.

*4.3. Comprehensive Evaluation of Audio-Visual Landscape*

The satisfaction scores of soundscapes, as rated by tourists, in each scenic location were standardized and placed in a two-dimensional quadrant with the standardized results of visualscape evaluation. Visualscape evaluation was represented on the horizontal axis, and soundscape satisfaction evaluation was represented on the vertical axis. The average value of the two evaluations was calculated as the dividing point of the horizontal and vertical axes to divide the four quadrants, and the final result was determined as the corresponding position in each of the four quadrants. Thus, we built a landscape audiovisual preference matrix model of the Koktokay GGp (Figure 6).

Quadrant I belongs to the "landscape key area" category, that is, the group of areas with the highest visualscape evaluation and soundscape satisfaction scores. As the landmark scenic location of the Koktokay GGp, Shenzhong Canyon is a typical natural geoheritage landscape with strong visual impact and high sensitivity. Soundscape resources are mainly natural sound sources, which are associated with high tourist satisfaction with their experiences. As the key area of perception and experience in geoparks, the protection of landscape resources can be continuously strengthened to ensure that the original landscape is not damaged, maintaining a benign development state in the long term.

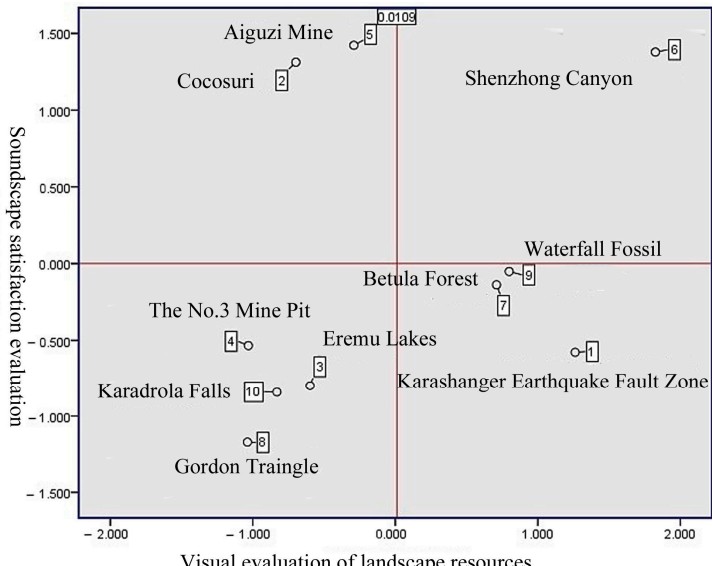

**Figure 6.** Landscape audiovisual preference matrix model of the Koktokay GGp. (1–10) Karashanger Earthquake Fault Zone, Cocosuri, Eremu Lakes, The No. 3 Mine Pit, Aiguzi Mine, Shenzhong Canyon, Betula Forest, Golden Triangle, Waterfall Fossil, and Karadrola Falls.

Quadrant II is the " landscape natural development area", i.e., the area with the lowest visualscape evaluation but high satisfaction with the soundscape. Located in this quadrant, the Aiguzi Mine and the Cocosuri scenic location contain both natural and humanistic sound sources, although they received low visual evaluation scores due to their special and unchangeable geographical locations. For this area, the uniqueness of scenic locations should be brought into play, and its irreplaceability within the geopark should be taken as the main development direction. The existing visual environment can be promoted by human means, such as by holding photography exhibitions to enhance the attractiveness of the scenery.

Quadrant III is the "landscape subpriority improvement area", comprising areas with low visualscape evaluation and soundscape satisfaction scores. The comprehensive visual evaluation of landscape resources was poor for the No. 3 Mine Pit, Eremu Lakes, Golden Triangle, and Karadrola Falls. The visual experiences of visitors to these scenic locations were not positive, in addition to lacking positive soundscape experiences. In view of the current development status, a reconsideration of concepts and innovation is required for the development of landscape resources. In particular, the No. 3 Mine Pit, as the landmark scenic location of the Koktokay GGp, requires innovative future development ideas and comprehensive transformation of its "field compatibility" to improve the tourist experience, both visually and acoustically.

Quadrant IV is the "landscape priority improvement area", i.e., areas with positive visualscape evaluations and low satisfaction scores with the soundscape. Located in this quadrant, Waterfall Fossil, Betula Forest, and the Karashanger Earthquake Fault Zone are mainly natural landscapes, and their comprehensive visual evaluation scores were high. In particular, as a world-class geological landscape in the Koktokay GGp, the Karashanger Earthquake Fault Zone has a high ornamental value. Therefore, it should be regarded as a strategic development, with a focus on improving its low SBE score through reasonable planning to give full play to the advantages of resources such as the structural characteristics of earthquake faults. Tourists can appreciate the beauty of the lines, structures, light, and shadow of the Earth's surface caused by previous earthquakes from various angles. However, tourists showed poor perception of the soundscape environment at this site. Therefore, soundscape elements that can enhance the atmosphere should be promoted, adjusting and protecting the rare acoustic environment through natural or artificial means to build the soundscape resource space while ensuring the authenticity of the scenic location.

## 5. Conclusions and Discussion

In this study, we comprehensively evaluated the landscape resources of the Koktokay GGp based on audiovisual factors. First, the SBE method was used to obtain a visual evaluation of professionals, particularly with respect to aesthetic attitudes. The SBE evaluation results show that visitors mainly rely on their own aesthetic evaluations, combined with the characteristics of the landscape itself, to reflect the actual aesthetic value of the landscape. However, for scenic locations that are entirely composed of geological phenomena, the SBE scores can sometimes be under-rated if visitors do not have enough basic geological knowledge while visiting.

Visualscape evaluation results were obtained by combining the GIS spatial analysis method with the SBE method. The research results show that sites such as Shenzhong Canyon, Karashanger Earthquake Fault Zone, Waterfall Fossil, and Betula Forest are advantageous scenic locations with rich visualscape resources. Our exploration of the influence of the physical properties of soundscape resources on tourist satisfaction revealed that tourists do not think that a higher or lower volume of a soundscape necessarily makes a location more comfortable and pleasant. Moreover, tourists' perceptions and experience of sound increase their own sound level thresholds with changes in environmental experience, which is consistent with the findings of Gang et al. [64]. In a study of urban park soundscapes, Liu et al. found that music-related sounds and water sounds had a positive impact on visitors' experiences, whereas traffic sounds had a negative impact [51]. Similar findings have been reported in other soundscape studies conducted in urban environments, i.e., that sounds related to nature, such as birds and plants, have a positive impact on satisfaction with urban landscapes, whereas sounds related to roads and traffic have a negative impact on satisfaction with urban landscapes [11]. The research findings related to the soundscapes of urban environments may provide reference significance for the soundscape management of geoparks. As a product of nature, the overall sound source composition of geoparks was originally mainly composed of natural attribute sounds, but the development of tourism activities may interfere with the sound environment within the scenic area. In the future, the addition of sound elements in harmony with the natural environment may improve tourists' experiences and perceptions.

This article contributes to the literature regarding the identification and evaluation of landscape resources. First, in this study, we comprehensively evaluated the landscape of a geopark in both the visual and auditory sensory dimensions, using IPA analysis to build an audio-visual preference matrix. This is a new perspective which clearly differs from previous studies on landscape resource evaluation that considered only a single dimension, echoing the call for multisensory tourism research with attention to the visual and auditory senses. The results presented herein can be used to optimize the content of traditional comprehensive evaluations of landscape resources and provide a scientific basis for the protection and development of GGps, as well as the promotion of their sustainable development.

Secondly, in this study, we combined the GIS spatial analysis method with the SBE method to a conduct a comprehensive evaluation of visualscapes from both subjective and objective perspectives. In contrast to previous studies that used only one landscape evaluation method [65,66], in this study, we attempted to combine the SBE method with GIS spatial analysis for the visualscape evaluation of the Koktokay GGp, as well as to integrate abstract statistical data with the subjective preferences of visitors for overall analysis, conforming to the current development direction of landscape evaluation. This was carried out in addition to evaluating landscape resources more scientifically, accurately, and comprehensively and improving the scientific evaluation results. In addition, the feasibility of applying the SBE-GIS comprehensive evaluation method to geopark landscape resource evaluation was verified, providing a reference for landscape resource evaluation in similar areas.

Third, in this article, we combined expert assessments and non-expert judgements. Existing research on visual landscape evaluation mainly focuses on expert assessments,

whereas in this study, we used the SBE method to obtain a subjective evaluation of visual landscape resources by professionals. At present, the two most common methods used in soundscape research are the combination of soundwalks with questionnaires and narrative interviews [67,68]. Based on the advantages of the questionnaire method and the nonlinear characteristics of geopark soundscape resources, in this study, we adopted the questionnaire method for objective non-expert evaluation of the geopark's soundscapes.

## 6. Limitations and Future Research

In this study, we used a comprehensive evaluation method to give full play to the advantages of multiple evaluation methods, eliminated the limitations of a single evaluation method, and made reasonable use of the advantages of each research method. However, the current study was subject to several limitations, which provide promising avenues for further studies. First, visualscape evaluation using photos and GIS spatial analysis resulted in a sample with a short time span, whereas the experience effect of a landscape varies with seasonal changes, as well as the light difference between the morning and evening, thereby affecting the evaluation results to some extent. Future researchers could use 3D, AR, and other technologies to analyze the appearances and surrounding environments of scenic locations in different seasons on the basis of this study. Secondly, the main focus of this study was a limited number of soundscapes and visualscapes, i.e., 10 typical scenic locations in the Koktokay GGp. Future research should be expanded to other kinds of destinations with different landscape resources. Thirdly, this study comprehensively utilized GIS spatial analysis, SBE, and the questionnaire survey method to evaluate the audio-visual landscapes of global geoparks. However, the underlying reasons behind the evaluation results need to be explored, and future research needs to further focus on studying why they exist.

**Author Contributions:** Conceptualization, Y.Z., C.Z. and X.P.; methodology, Y.Z. and X.P.; software, Y.Z.; validation, Y.Z.; formal analysis, Y.Z. and X.P.; investigation, Y.Z., C.Z. and X.P.; data curation, Y.Z.; writing—original draft preparation, Y.Z. and X.P.; writing—review and editing, C.Z. and X.P.; visualization, Y.Z.; supervision, C.Z.; funding acquisition, Y.Z. and C.Z. All authors have read and agreed to the published version of the manuscript.

**Funding:** This research was funded by Innovation Project of Xinjiang Uygur Autonomous Region Graduate Education (Study on Sports Addiction of Ski Tourists), grant number XJ2023G080; Start-up Fund for Doctoral Research of Xinjiang University, grant number BS202202 and Xinjiang Uygur Autonomous Region Social Science Foundation Project, grant number 2023BGL073.

**Data Availability Statement:** The data presented in this study are available upon request from the corresponding author. The data are not publicly available due to data publisher regulations.

**Acknowledgments:** Thanks to all editors and reviewers.

**Conflicts of Interest:** The authors declare no conflict of interest.

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
