# Peer review of "Key Area Recognition and Evaluation of Audio-Visual Landscape for Global Geoparks: A Case Study of Koktokay in China"

_land, doi:10.3390/land12091799_

Round 1

Reviewer 1 Report

In general, it is a well-structured work and the purpose of it is understood. Very good work.

Below, I write some observations so that they can be considered by the authors:

I am not very acquainted to the methods used for the visual evaluation of the landscapes considered by the authors, however the level of detail and synthesis of the work made me understand every detail of this manuscript.

It is interesting to observe that, according to this work, the experience of a visitor is not influenced by the absence or presence of sounds, therefore the sense of satisfaction is related to a series of variables that can be covered in subsequent studies.

It may be interesting that the authors have made a comparison of the results of scenic beauty evaluation and evaluation of soundscape, that is, if the evaluations of each variable keep some level of correlation. In turn, it may be interesting for tourists to do both evaluations (SBE and evaluation of soundscape), this may be a future work. I don't know if the fact that people who are knowledgeable in landscapes evaluate the visual conditions, and tourists evaluate the perception of sounds, can generate different interpretations. Even so, the work is very interesting and entertaining.

Specifically, I have made some specific observations, for the consideration of the authors:

Line 36: Correct this information, geoparks have not existed since 2015, but long before, direct recognition by UNESCO is from that date. Please in the text specify this not minor detail.

Line 54: You can also add to these correct statements that it is necessary to establish public policies for the sustainability and conservation of resources, in addition to the aforementioned.

Line 135: Shouldn't the bibliographic citation go next to Daniel (...)?

Line 171: Is it possible to add a bibliographic citation here?

Line 177: It is possible to add the Geopark area, to have a better spatial relationship.

Line 185 to 190: In the text the authors mention ten typical scenic locations that best represent the landscape resources of the geopark. These sites correspond to geosites of the geopark? Or are they non-inventoried resources? Please define this in the text.

Line 295: I recommend placing this information after the description of figure 3 (line 297).

Line 310: I recommend placing this information after the description of figure 4 (line 312).

Line 324: I recommend placing this information after the description of figure 4 (line 326).

Author Response

Dear reviewer,

We would like to thank you for your time and effort in reviewing our paper and providing constructive comments. Those comments are all valuable and very helpful for revising and improving our manuscript, as well as the important guiding significance to our researches. We have studied the comments carefully and the manuscript has been revised carefully and strictly according to your comments, which we hope meet with approval.

We are here resubmitting the revised manuscript entitled “Key Area Recognition and Evaluation of Audio-visual Landscape for Global Geoparks: A Case Study of Koktokay in China” for your kind consideration of its suitability for publication in Land. In order to facilitate your review, bold and marked red fonts were used to show the corresponding response. And we have also updated our manuscript by properly adding these responses into the revised version. "Please see the attachment." 

Our deepest gratitude goes to you for your careful work and thoughtful suggestions that have helped improve this paper substantially.

Best Regards,

Authors

Reviewer 2 Report

The article presents the results of the questionnaires, measuring, based on the Likert scale, levels of visual and soundscape satisfaction in the selected points in Koktokay Geopark in China. Results are presented in a very elaborate manner.

However, based on these results, it is not possible to make recommendations for installing artificial sounds of music or birdsong, etc., in the geopark, where, probably, the main value of visiting is experiencing the original nature and artificial and fake elements could disturb the experience. This can be a subject of discussion, but not a suggestion (line 456-458), and certainly not based on the results of this research.

The results only show that places rated poorly will require additional research in the future on why it is so.

Author Response

(The authors gave the same response as above.)

Reviewer 3 Report

Comments and suggestions for authors

Dear Authors,

The article is informative and well structured. However, I have some questions:

1) Is the equation for calculation developed by the authors? If there are authors of the methodology, then a link is needed. 214, 235, 243 lines...

2) Likert scale was used in the survey results, but I did not find any analyses in the work. 213, 241 lines

3) Which questionnaires were filled out by tourists? Are there any survey results? line 239

4) In line 258 there is such a phrase: "The scores of the 50 evaluators were calculated to derive the SBE score and the standardized score of each scenic location as follows (Table 2)".  Who are the "50 evaluators"? Tourists or working in the field of tourism?

Author Response

(The authors gave the same response as above.)
